



# Bioaerosols as indicators of central Arctic ice nucleating particle sources

Kevin R. Barry[1,*], Thomas C. J. Hill[1], Sonia M. Kreidenweis[1], Paul J. DeMott[1], Yutaka Tobo[2,3] and Jessie M. Creamean[1]

[1]Department of Atmospheric Science, Colorado State University, 1371 Campus Delivery, Fort Collins, Colorado 80523-1371, United States of America
[2]National Institute of Polar Research, Tachikawa, Tokyo 190-8518, Japan
[3]Graduate Institute for Advanced Studies, SOKENDAI, Tachikawa, Tokyo 190-8518, Japan

*Correspondence to: Kevin R. Barry (Kevin.Barry@colostate.edu)

**Abstract.** The Arctic is warming at a rapid rate, with implications for microbial communities as the ecosystems change. Some microbes and biogenic materials can affect the persistence of long-lived mixed-phase clouds by serving as ice nucleating particles (INPs). The presence of INPs modulates the cloud phase, and long-term measurements are important to elucidate their seasonal sources and predict future change. The Multidisciplinary drifting Observatory for the Study of Arctic Climate (MOSAiC) expedition in 2019-2020 provided the first year-long measurements of bioaerosols and INPs in the central Arctic. Here, we investigated the INP seasonal cycle and its relation to the seasonal cycle of bacteria and eukaryotes. INPs were greatly elevated and compositionally similar in summer, aligning with a greater prevalence of local bioaerosol sources, but despite this, a diverse mixture of sources (marine and terrestrial) was present all times. A common broader Arctic INP population is hypothesized for much of the year by comparable coincident data collected in Svalbard and a sensitivity of both the INPs and bioaerosols to large-scale events.

## 1 Introduction

Arctic surface air temperatures are warming three or four times faster than the rest of the world from Arctic amplification (Rantanen et al., 2022; Zhou et al., 2024). This surface warming affects Arctic ecosystems, including microbial communities. Land effects include: greening of tundra vegetation (Berner et al., 2020); thawing of permafrost containing actively transcribing microbes (Wu et al., 2022); and warming of soil, which can alter the frequency of bacterial taxa (Newsham et al., 2022). Ocean effects include increased Atlantic water influx, which could replace cold-adapted bacteria (Carter-Gates et al., 2020); increased sources of dissolved organic matter that could alter bacterial abundance (Nguyen et al., 2022); increased light from sea-ice loss resulting in earlier onset of primary production (Lannuzel et al., 2020); and ocean acidification (Gamberg, 2020).

The effects of changing Arctic ecosystems extend to the atmosphere through microbial emissions from the surface. Some aerosols from these sources can affect clouds through serving as ice nucleating particles (INPs). INPs trigger ice



formation at temperatures warmer than homogenous freezing (-38 °C), and have many sources, including bacteria, fungi, organics, and mineral dust (Hill et al., 2018; Kanji et al., 2017; Murray et al., 2012). The lifetime, thickness and phase of Arctic

clouds affect the level of Arctic amplification through impacting the surface energy budget (Tan & Storelvmo, 2019). Arctic mixed-phase clouds occur about 40% of the year, and most have temperatures between -25 and -5 °C, a range impacted by many INP sources (Shupe et al., 2006). Modeling studies have shown the treatment of INPs influences the strength of the cloud phase feedback (Tan et al., 2022).

Some aspects of the Arctic aerosol annual cycle are understood, particularly the transport of anthropogenic emissions

from lower latitudes that typically occurs from January to April and is responsible for the Arctic haze phenomenon. In contrast, the summer aerosol, between June and September, is characterized by increased local influence (Schmale et al., 2021) and tends to be dominated by poorly-quantified natural sources.

Some local sources of aerosols, enriched with biogenic INPs, include glacial soil dust and leaf litter (Barr et al., 2023; Conen et al., 2016; Tobo et al., 2019), and marine sources, especially near phytoplankton blooms (Creamean et al., 2019;

Hartmann et al., 2021; Wilson et al., 2015). Thermokarst regions contain particularly active, unrepresented potential sources of Arctic INPs (Barry et al., 2023b; Creamean et al., 2020). Recent modeling studies highlighted Arctic dust as a potential important contributor of INPs (Kawai et al., 2023; Shi et al., 2022), as glacial soil dust accounted for nearly all dust INPs in the Arctic lower troposphere between June and November (Kawai et al., 2023).

Long-term Arctic INP measurements reporting the annual cycle have mostly occurred at fixed coastal sites (Pereira

Freitas et al., 2023; Sze et al., 2023; Tobo et al., 2024; Wex et al., 2019). They detected seasonality in INP concentrations, with INPs active at warmer temperatures (>-15 °C ) in highest abundance in summer. Higher concentrations of airborne bacterial cells and fungal spores were also found in summer (Abrego et al., 2024; Johansen, 1991; Johansen & Hafsten, 1988; Šantl-Temkiv et al., 2019), with an increase in potential local sources (Jensen et al., 2022). However, in the central Arctic over the pack ice, INP measurements have been limited to specific months (Bigg, 1996; Bigg & Leck, 2001; Hartmann et al., 2021;

Porter et al., 2022). The Multidisciplinary drifting Observatory for the Study of Arctic Climate (MOSAiC) campaign campaign on the *R/V Polarstern* provided the first year of aerosol INP and bioaerosol measurements in the central Arctic (Creamean et al., 2022). To identify contributors, link source samples to potentially the most active INPs, and as a proxy of airmass origin, we report the first annual cycle of central Arctic aerosol bacteria and eukaryotes.

## 2 Methods

**2.1 Sample collection during the MOSAiC expedition**

The MOSAiC expedition took place from October 2019-September 2020 in the central Arctic aboard the German Research Vessell (R/V) *Polarstern*, separated into 5 legs. The vessel drifted passively in ice between: October 4-December 13 (Leg 1); December 13-February 24 (Leg 2); February 24-May 16 (Leg 3); June 19-July 31 (Leg 4); and August 21-September 20 (Leg 5). The other periods were time when the ship was in transit. Overviews of this expedition are in Nicolaus et al. (2022),

Rabe et al. (2022), and Shupe et al. (2022).



Aerosols for DNA and INP analyses were collected on *Polarstern's* P-deck, using filter samplers mounted about 15 m above sea level with the U.S. Department of Energy Atmospheric Radiation Measurement (DOE ARM) AMF2 facility. For DNA analyses, polycarbonate filters (0.4 µm Whatman Nuclepore track-etched hydrophilic membranes) were precleaned by soaking in 10% H2O2 followed by 0.1 µm filtered deionized (DI) water rinses (Uetake et al., 2020). Polycarbonate filters for INP analyses (0.2 µm Whatman Nuclepore track-etched hydrophilic membranes) were precleaned by brief ultrasonication in methanol followed by 0.1 µm filtered DI water rinses (Barry et al., 2021). Filters for both DNA and INP analyses were precleaned and preloaded in Nalgene units in a laminar flow cabinet at Colorado State University (CSU). Identically cleaned 10 µm polycarbonate filters were loaded underneath the 0.2 and 0.4 µm filters to provide a clean support for the sample filter. Filters were typically collected for 3-day periods, with an average total volume of air filtered of 139,500 standard Liters (sL: 0 °C; 1013.25 mb) for DNA filters and 88,800 sL for INP filters.

Samples of seawater, sea ice, snow, melt pond water, and open lead ice were collected and used to identify potential local sources of biological aerosols. Source sample metadata, including types, collection dates and times, latitudes/longitudes, and depths are provided in Table S1. Seawater includes samples from *Polarstern's* flowthrough seawater tap system (FT) collected at 11 m depth and from a CTD (conductivity, temperature, depth) rosette at 4-7 depths within the upper 400 m. Sea ice cores were collected using a Kovacs II coring system. Ice cores were sectioned into 5-10 cm segments, melted, then diluted with 0.22 µm filtered seawater. Snow samples were collected from the surface, middle, and bottom of the snow pits. Melt pond and newly formed lead ice samples were collected mainly during the summer months. Protocols are further described in Nicolaus et al. (2022) and Rabe et al. (2022). All filters and samples were stored at -20 °C for the duration of the campaign, during transport, and at CSU until analysis.

**2.2 DNA sample analysis**

We processed aerosol filters and source samples for 16S rRNA gene (bacteria). A subset of aerosol filters and source samples were processed for ITS (fungi) and aerosol filters for 18S rRNA gene for eukaryotic composition. These samples were processed similarly to previous work in our lab (Barry et al., 2023a; Uetake et al., 2020).

aerosol filter samples were extracted. The processing of 64 samples followed the standard extraction protocol, cutting up the filter in pieces, 30 s ultrasonication in 2 mL of nuclease free water, and concentration with a Microcon DNA Fast Flow Centrifugal Filter. Extraction was done with the DNeasy PowerLyzer Microbial Kit (Qiagen).

A different extraction protocol was employed for 79 source samples and 7 additional aerosol filters processed later. Ice, seawater, melt pond, flowthrough, and snowmelt filters were pre-processed identically, by thawing the samples and filtering approximately 30 mL through a Sterivex (Millipore) 0.22 µm pore filtering unit (~15 mL for snow). The Sterivex was separated with a PVC pipe cutter (Cruaud et al., 2017), and the filter detached with a sterile scalpel and cut into pieces. For the water and additional aerosol samples, filter pieces were placed directly into the extraction tubes of the DNeasy PowerSoil Pro Kit (Qiagen). This kit and modified method were chosen to remove the concentration pre-step where losses may occur, and to allow extraction from the filter to proceed directly. Extraction for both batches followed their respective Qiagen protocol, with two elutions used in the final step to improve recovery.



All aerosol and source samples were amplified for 16S rRNA. The V4-V5 region was targeted with the 515yF/926pfR primers (Parada et al., 2016) with cycling conditions following (Uetake et al., 2020) and the UCP Multiplex PCR master mix (Qiagen). 29 aerosol and 36 source samples were amplified for ITS. The primers followed Walters et al. (2016), and cycling conditions were: 95 °C for 2 min; 37 cycles of 95 °C for 30 s, 55 °C for 60 s, 72 °C for 60 s; followed by a 72 °C hold for 5 min. 29 aerosol samples were amplified for 18S rRNA. We used the Euk1391f-EukBr primer pair detailed on the Earth

Microbiome Project (2017). Cycling conditions were also adapted from the Earth Microbiome Project: 94 °C for 3 min; 37 cycles of 94 °C for 45 s, 57 °C for 60 s, 72 °C for 90 s; followed by a 72 °C hold for 10 min.

The primers contained Illumina adapters and were purified with AMPure XP (Beckman Coulter) two times: after the 1st amplification and 2nd amplification that added sample barcodes (IDT for Illumina Nextera DNA UD Indexes). This 2nd PCR step had cycling conditions of 95 °C for 5 min; 12 cycles of 95 °C for 30 s, 60 °C for 30 s, and 72 °C for 30 s; followed

by a 72 °C for 7 min. This step used the AmpliTaq Gold LD DNA Polymerase (Applied Biosystems). After 2nd purification, samples were quantified with the Quant-iT™ 1X dsDNA Assay Kits (Invitrogen) on an Enspire plate reader, to create an equimolar library. This library was sequenced at the CSU Next Generation Sequencing Core with the Illumina MiSeq Reagent Kit v3 (600-cycle). Source samples for ITS were sequenced later, and so the sample pool was prepared at CSU identically to the prior samples but sequenced at RTL Genomics (Lubbock, TX) using the same sequencing kit.

Next, sequences were demultiplexed in the Illumina BaseSpace Sequence Hub, before being imported into QIIME2 Version 2024.5 for processing (Bolyen et al., 2019). Reads were denoised with DADA2 (Callahan et al., 2016) to create an amplicon sequence variant (ASV) table. For 16S, preformatted reference sequence and taxonomy files were based on SILVA 138 (Quast et al., 2012; Robeson et al., 2020). For 18S, reference sequence and taxonomy files were from the PR2 (Protist Ribosomal Reference) database, version 5.0.0 (Guillou et al., 2012; Vaulot et al., 2023). For ITS, reference sequence and

taxonomy files were from UNITE Community database (all eukaryotes), version 10.0 (Nilsson et al., 2019). Taxonomy assignment used the feature-classifier plugin in QIIME2 (Bokulich et al., 2018; Pedregosa et al., 2011). For 16S, any non-bacterial reads were removed (mitochondria, chloroplast, archaea).

We utilized several field and laboratory controls. Blank aerosol filters were prepared and handled identically as the samples, minus airflow. For water samples, we put 30 mL of nuclease free water into a 50 mL centrifuge tube and through a

Sterivex unit before extraction. Additionally, several extraction and PCR negatives were included. In total, for 16S: 3 aerosol, 4 extraction, 2 Sterivex, and 5 PCR; ITS: 1 aerosol, 1 extraction, 1 Sterivex, and 1 PCR; 18S: 1 aerosol, 2 extraction, and 2 PCR controls were done. An example of the negative and positive control taxonomy (from the main aerosol filter sequencing run) is given in Figure 1. The positive control used was ZymoBIOMICS® Microbial Community Standard, and shows that we were able to detect gram negative and positive bacteria in similar percentages.

Since contamination potential is high from the ship and laboratory environmnet, we did a multi-component blank correction approach for the 16S rRNA aerosol filters. For these, ProkAtlas (Mise & Iwasaki, 2020) was used to assign potential human contamination and is detailed below for enviornmental use. For blank correction, any ASVs that had more than 50% attribution from the human environment were excluded. Next, the decontam package prevalence method with a threshold of



0.75 was used (Davis et al., 2018). For 16S rRNA source, 18S rRNA, and ITS samples, only the decontam step was used with
a threshold of 0.5 for the prevalence method.

For 16S rRNA, 47 aerosol filters and 77 source samples had more than 1000 reads post-blank correction and were
used for downstream analyses. For ITS aerosol samples, only 10 samples amplified, but those amplified well (minimum of
41287 reads after blank correction). For the ITS source samples, 27 were used for downstream analyses (minimum of 2081
reads after blank correction). For 18S aerosol samples, 24 were used for downstream analyses (minimum of 6268 reads after
blank correction).

Analysis methods performed for the 16S rRNA data include sample pooling, source attribution, and alpha diversity.
To better represent the bacterial seasonal cycle, the aerosol filters were pooled by month of filter start date for taxonomic plots,
using the "mean-ceiling" function to combine the frequencies of identical ASVs. There was only one sample in October (start
date of 10/27/2019), which was included with November. For alpha diversity analysis, we rarefied at 2269 reads. Source
attribution  assigned reads to potential environments through ProkAtlas (Mise & Iwasaki, 2020). Categories were collapsed by
summing: marine and seawater for "Marine"; freshwater, glacier, lake water, wastewater and aquifer for "Freshwater"; soil,
phyllosphere, rhizosphere, plant, fungus, and terrestrial for "Terrestrial"; peat, groundwater, rice paddy, sediment, permafrost,
freshwater sediment, and marine sedimentfor "Sediment"; aquatic, salt marsh, hypersaline lake, and estuary for "Other Water";
gut, insect, insect gut, mouse gut, feces, termite gut, pig gut, rat gut, chicken gut, and bovine gut for "Animal Association."

Analysis methods performed for the ITS data included SourceTracker2 (Knights et al., 2011). We rarefied at 17029
reads, which retained 26 source and 10 aerosol samples. We composited the samples into three categories (Snow, Seawater,
and Ice and Melt Pond).

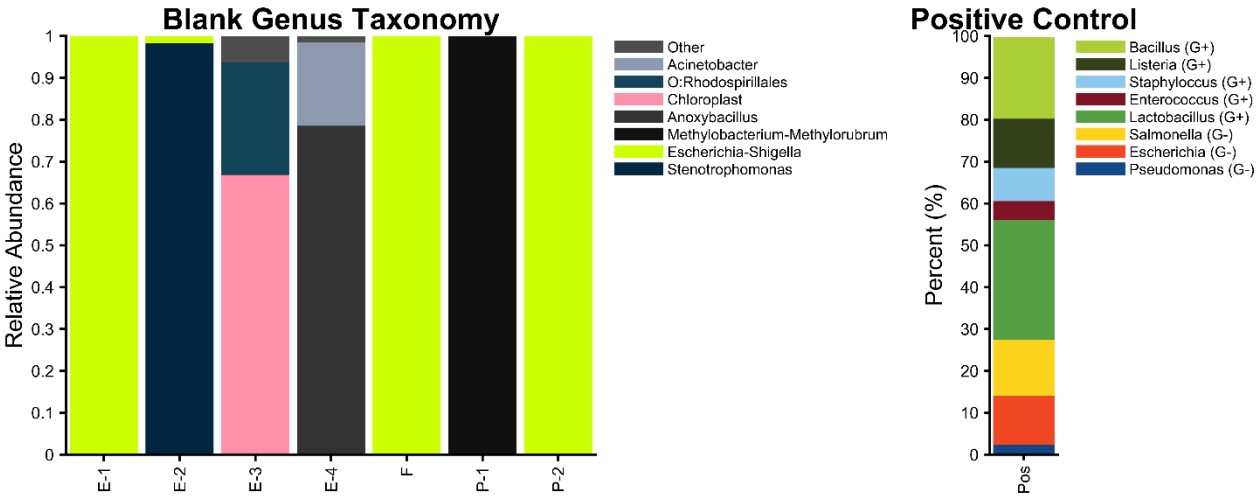

**Figure 1:** (Left): 16S blank relative abundace (genus) for 4 extraction controls (E-1,E-2,E-3,E-4), 1 aerosol filter control (F),
and two PCR controls (P-1,P-2). (Right): 16S positive control percentage of sequenced reads.



### 2.3 INP sample analysis

For INP processing of 73 filters, 8 mL of 0.1 µm filtered DI water were added to a filter in a prerinsed 50 mL centrifuge tube to create a suspension and shaken for 20 min in a Roto-Torque rotater (Cole Parmer). For each, 11-fold dilutions were made (400 µL sample and 4000 µL 0.1 µm filtered DI water) and pipetted out in 32 50 µL aliquots into PCR trays (Optimum Ultra). A 32 50 µL block of 0.1 µm filtered DI water was included with each sample as a negative control. The PCR trays were placed into the aluminum blocks of the CSU Ice Spectrometer (IS) and cooled at 0.33 °C min$^{-1}$. The current setup of the IS—with conversion of data to equivalent atmospheric loading (INPs sL$^{-1}$ of air) and blank corrections—is detailed in DeMott et al. (2018). Four field blanks were transported and processed identically (without airflow) and combined to correct the samples by using an average regression. Blank corrections had virtually no effect on INP concentrations, as there was only an average of 11 INPs per blank filter at -25 °C, while sample filters typically had >1000 INPs at this temperature. Thermal and chemical treatments were performed on 26 of the remaining suspensions. These treatments have been used extensively to infer INP composition (Barry et al., 2023a; Hill et al., 2016; McCluskey et al., 2018; Suski et al., 2018; Testa et al., 2021). Heat treatment at 95 °C removes heat labile INPs (such as proteins), and hydrogen peroxide ($H_2O_2$) digestion at 95 °C removes all organics. Thereby, heat labile and heat stable organic INP fractions can be derived, with the remaining inorganic (presumed mineral).

Weekly INP data obtained at Zeppelin Observatory (78.91° N, 11.89° E) in Svalbard (Pereira Freitas et al., 2023; Tobo et al., 2024) are included for comparison to the data collected on the Polarstern during MOSAiC. The INP data at Zeppelin Observatory were analyzed with the Cryogenic Refrigerator Applied to Freezing Test (CRAFT) system (Tobo, 2016), which uses a cold plate isolated in a clean room. Methods previously compared well with the IS (DeMott et al., 2017) .

### 3 Results and Discussion

### 3.1 Seasonal cycle of INP composition

First, we present the seasonal cycle of INP composition for MOSAiC, partially presented in Creamean et al. (2022). Based upon heating and $H_2O_2$ digestion, we subdivided sample INP populations into heat labile organics (presumably biological), heat stable organics, and inorganic (Fig. 2). The summer samples were compositionally similar, as June through August samples had virtually 100% heat labile organic INPs active at both -15 and -20 °C, and >80% at -25 °C (except August 2). Inorganic influence was largest during winter and at the coldest temperatures, which broadly aligns with the Arctic haze period where dust may also be transported from lower latitudes (Schmale et al., 2022). However, during all seasons, organic INPs contribute down to at least -25 °C and are therefore present in the range of mixed-phase clouds.

The heat labile maximum in summer is reflective of enhanced biological productivity sea ice mimimum, glacial retreat, and less snow coverage. This finding also agrees with recent work showing heat labile fractions of over 90% in summer and 50-85% in winter at -12 °C at Zeppelin Observatory (Pereira Freitas et al., 2023), and 100% above -20 °C near the North Pole in August and September (Porter et al., 2022).




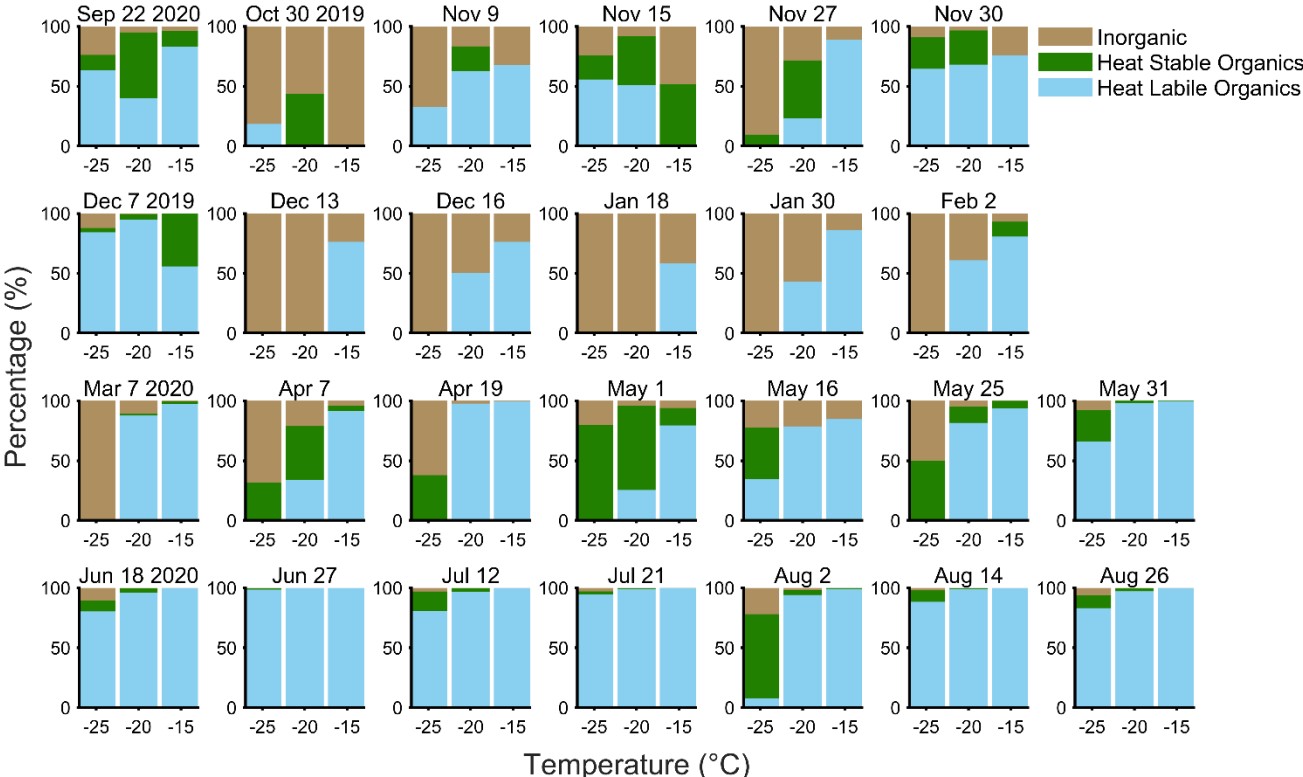

**Figure 2:** Seasonal cycle of the INP composition in the temperature regime at -15, -20, and -25 °C measured on the R/V *Polarstern* during MOSAiC. Date refers to the aerosol filter start date, with blue denoting the heat labile organic fraction (from heating to 95 °C), green denotes the heat stable organic fraction (from digestion with $H_2O_2$), and tan is the inorganic (presumed mineral) contribution from the INPs remaining after $H_2O_2$ digestion. Additional information on the seasonal cycle of INP abundances is in the Supplement (Fig. S1).

### 3.2 Seasonal cycle of bioaerosols

The Arctic annual cycle of bacterial taxa in ambient aerosol indicates a diverse and complex community (Fig. S2). The psychrophile, *Polaribacter*, was the most abundant genus overall, 10% in aggregate and detected in 5 months, peaking in summer and autumn. This taxon has been found in high abundance in seawater samples after spring phytoplankton blooms in the North Sea (Teeling et al., 2016), and in summer aerosol over the Southern Ocean (Uetake et al., 2020). Among the other MOSAiC aerosol top-20 genera, *Sphingomonas*, *Hymenobacter*, and *Methylobacterium* (Fig. S2), all widely distributed in nature, were previously detected in high relative abundances in air from Station Nord, Greenland, between March and June (Tignat-Perrier et al., 2019). Surprisingly, Figure S3 shows that bioaerosols had almost no overlap with the major taxa identified in the source samples, suggesting the main bioaerosol sources were not well represented in the source samples and could be non-local.

Next, we attributed each of our aerosol sequences to their most likely environmental origin to obtain their fractional source contribution (Fig. 3). Potential bacterial contributions attributed to freshwater, sediment, and animals were found in similar proportions throughout the year. The aerosol contained the largest marine influence during summer (52% in August), and considerable terrestrial influence in all months (>15%). The alpha diversity of the aerosol samples wasn't signficantly (p<0.05) different between seasons (Fig. S4), despite increased variability in the spring and summer, providing further evidence that the bioaerosols were diverse taxonomic mixtures. Marine influence was at a minimum during the winter and spring, despite consistent freshwater contributions, and when combined with persistent terrestrial influence, suggests the bacteria came from longer-range sources as Arctic freshwater sources are frozen during this time.

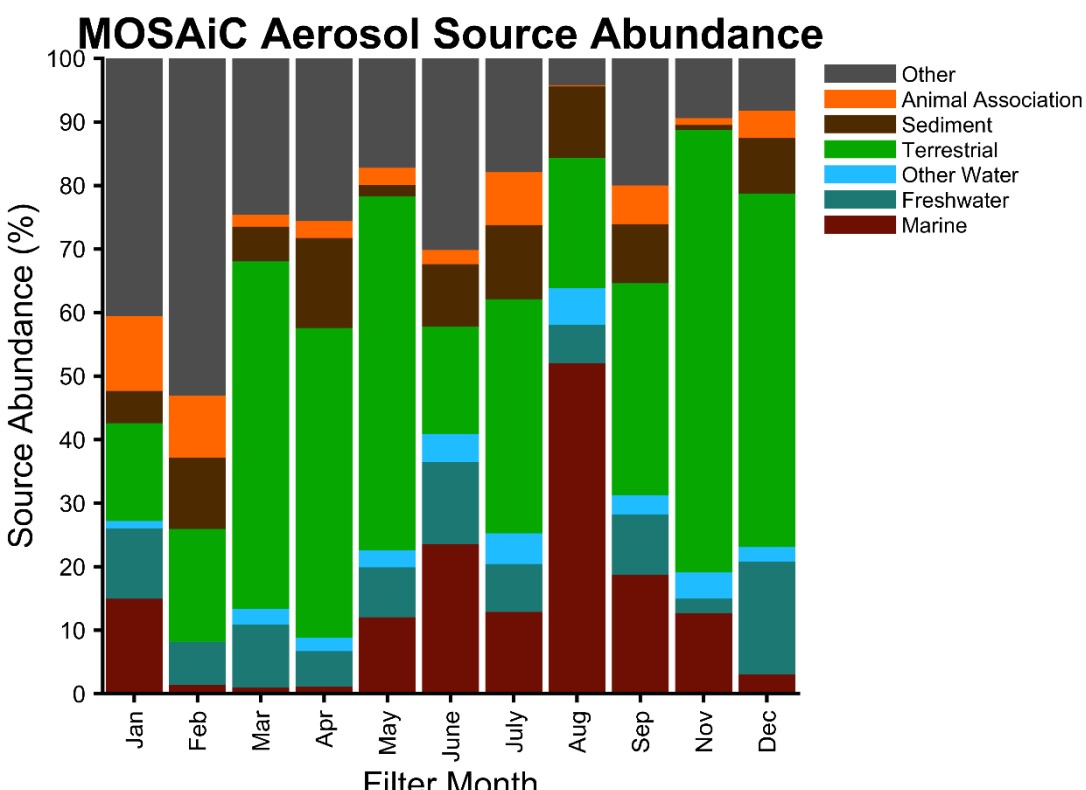

**Figure 3:** Potential bacterial source attribution (percentage) based on typical habitat for the pooled aerosol samples as a function of month, combined from ProkAtlas (Mise & Iwasaki, 2020).

The eukaryotic aerosol annual cycle also revealed complex seasonality (Fig. S5). Source tracking of the ITS data (Fig. 4) identified some influence of local sources, especially in the ice and melt pond category during mid summer. Snow also had ASVs detected in 4 aerosol samples. The ASVs common to the aerosol and ice/melt pond samples may be attributed to *Cryolevonia,* which has been isolated from permafrost in the Alps and sea ice (De Garcia et al., 2020; Pontes et al., 2020). These ASVs were previously assigned to this genera in the UNITE 9.0 database, but with 10.0, are only resolved to the class

Microbotryomycetes, which contains *Cryolevonia* (Fig. S5). The sequences common to both the aerosol and local sources may have also originated from more distant Arctic sources, such as an amalgamation of melt ponds (maximum coverage on June
30: (Wang et al., 2024)), or re-suspension from previous deposition as the ice near the *Polarstern* contained Siberian sediment (Krumpen et al., 2020). Nonetheless, sequences in the source and aerosol samples indicate that these categories have the potential to contribute to the bioaerosol and heat labile INPs.

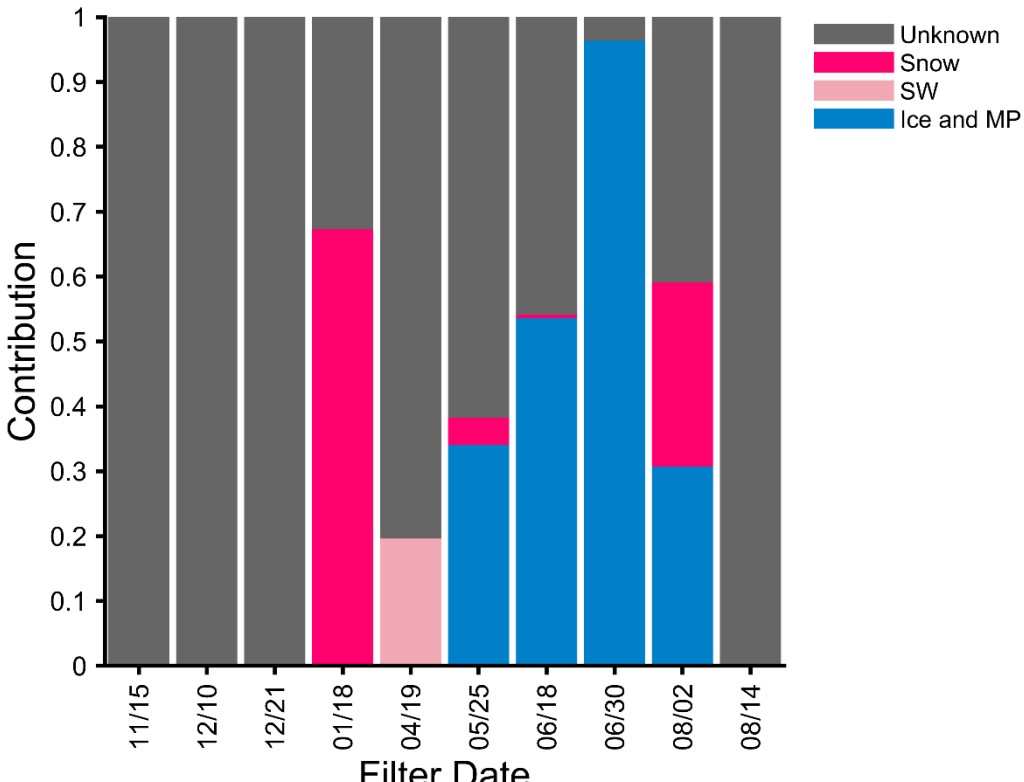

**Figure 4:** ITS source tracking analysis for 10 aerosol samples, organized by month of the campaign (November 2019-August
2020). The listed dates refer to the start date of the filter. 28 source samples were included and composited into three categories: Ice and Melt Pond (MP: Blue); Seawater (SW: Light Pink); and Snow (Dark Pink).

The 18S rRNA gene results show the eukaryotic bioaerosol was dominated by fungi, primarily Basidiomycota and Ascomycota (Fig. 5). Excluding fungi, Bacillariophyceae (diatoms) and Mollusca (mollusks) were observed in multiple spring
and summer samples, consistent with increased marine bacterial taxa. Previously, continental work showed a higher normalized species richness of Ascomycota in the winter and spring (Fröhlich-Nowoisky et al., 2009), and at greater relative proportions in marine/coastal air (Fröhlich-Nowoisky et al., 2012), due to the smaller spore size of Ascomycota. We generally found increased relative abundance of Basidiomycota in summer, however the seasonal trends are somewhat ambiguous.



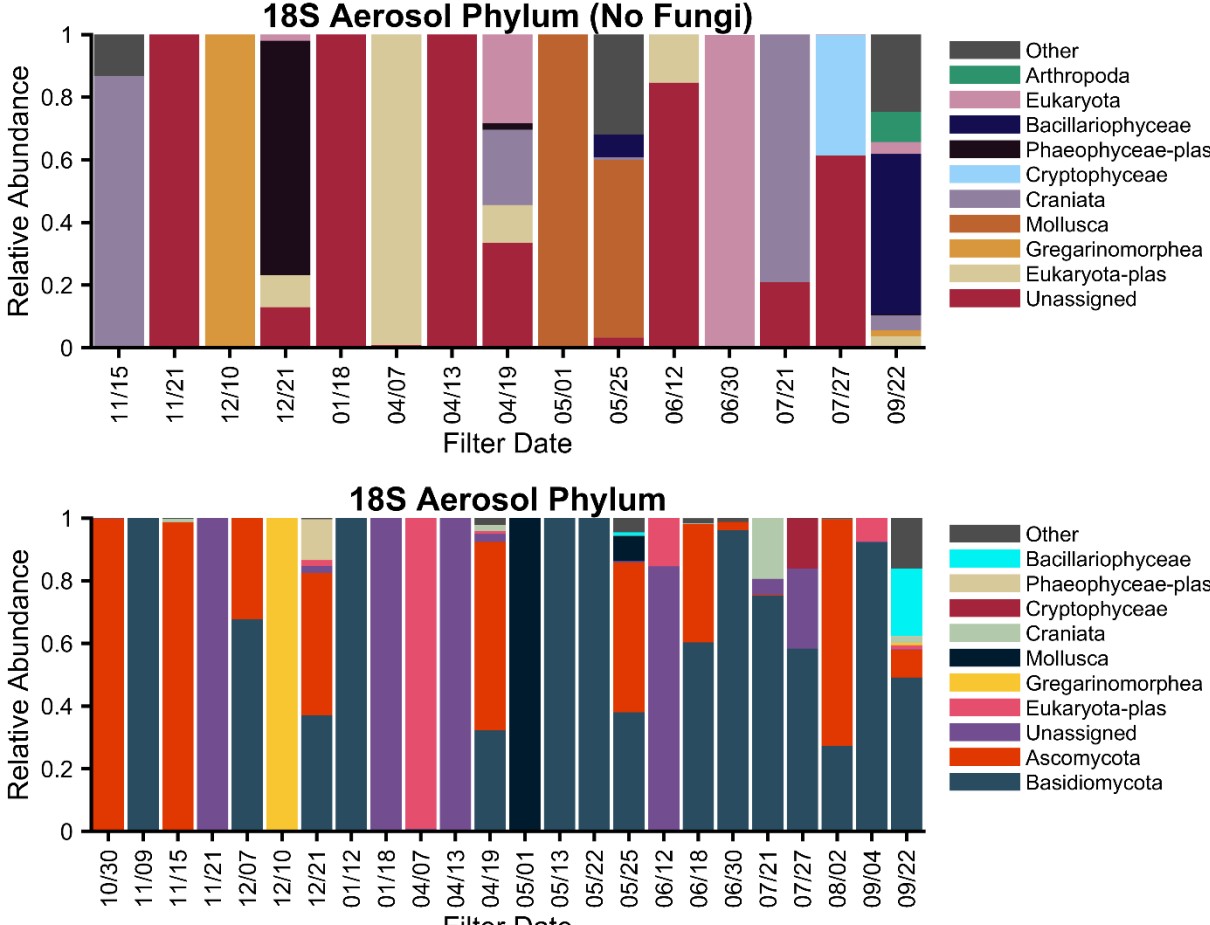

**Figure 5:** Relative abundance taxonomy for the MOSAiC aerosol for 18S at the phylum level, excluding fungi (top), and with fungi (Ascomycota and Basidiomycota, bottom). The filter starting date is given on the x-axis, and samples are colored by the top 10 abundant taxa.

### 3.3 Investigating potential INP origins through bioaerosol linkage

To investigate bioaerosol and heat-labile INP airmass origins, INPs at -15 °C and biological tracers were plotted with airmass resident time percentages along 5-day HYSPLIT back trajectories only using points ≤ 500 m above mean sea level (Fig. 6). This yields the percentage of time spent over the indicated surface type (Creamean et al., 2022). Ice is greater than 85% coverage, marginal ice zone (MIZ) is 15-85% coverage, and ocean is <15% coverage. Based upon their presence in source samples, we assume the ASVs resolved to Microbotryomycetes as a tracer for local/regional Arctic sources; *Polaribacter*, a cold-dwelling marine bacteria, as a tracer for local/regional marine airmasses; and 18S fungal presence as a general terrestrial marker. Although marine fungi are present in the Arctic (e.g. Hassett et al., 2019), their isolation sources from the Basic Local





Alignment Search Tool tool in Geneious Prime 2023.2.1 indicates the overwhelming majority of our 18S fungal ASVs were terrestrial. Transport of fungal spores from terrestrial sources in summer has been hypothesized, with enhanced concentrations of fluorescent aerosol particles ascribed to emissions from the tundra (Pereira Freitas et al., 2023; Perring et al., 2023).

**Figure 6:** (Top) INP concentration at -15 °C and corresponding presence (plotted around 1)/absence (plotted around 0) of one fungal taxon (Microbotryomycetes: maroon) and one bacterial taxon (*Polaribacter*: blue). The presence/absence of fungi is also indicated with purple squares (from the 18S data). (Bottom) Percent contribution to 5 day <500 m back-trajectory and corresponding position of the ship. Ice (green) refers to greater than 85% sea ice concentration (SIC), MIZ (dark blue) is the marginal ice zone and refers to 15-85% SIC, Ocean (blue) refers to the ice-free ocean <15% SIC, and Land (brown). Gray boxes indicate the November storm (11/16-20/2019) and the April warm air mass intrusion (4/15-21/2020).

The seasonal cycle of warm-temperature INPs is evident in Figure 6, increasing by orders of magnitude from winter to summer. When the highest INP concentrations were observed in summer, the co-occurrence of the regional biological tracers increased. A mixed airmass contribution was usually coincident with the presence of tracers. For example, for the May 25-28



filter, which had daily airmass contribution maxima of 83% ocean, 77% ice, and 5% land, we detected Microbotryomycetes, *Polaribacter*, and fungi from 18S. Although local sources contributed to the summer aerosol (Fig. 4), these contributions were unlikely to be from a single environment (e.g. solely marine or terrestrial), and regional transport cannot be ruled out.

While INPs followed clear seasonality, their concentrations were affected by periodic large-scale events. Warm air mass intrusions (WAMIs) from cyclones in November 16-20, 2019, and April 15-21, 2020 are shown in gray shading (Fig. 6). 18S fungi and *Polaribacter*, but not Microbotryomycetes, were detected during the November storm (Rinke et al., 2021), indicating a mixture of aerosol sources. Back trajectories indicated a shift from air predominantly over the ocean on November 16, to predominantly over the ice on November 17-20. *Polaribacter* was not detected until the November 18-21 filter (not sequenced for fungi), so this airmass transition could be responsible for different bioaerosol populations. Concurrently, INPs

at -15 °C increased by an order of magnitude from $1.7 \times 10^{-4}$ to $1.8 \times 10^{-3}$ L$^{-1}$, with over a 6-degree warmer freezing onset temperature. After November 18-21, *Polaribacter* was not detected again until May 2020, and higher INP concentrations at -15 °C were not detected until January.

During the April 15-21 WAMI (Dada et al., 2022), the INPs were greatly elevated compared to the periods before and after, reaching 0.12 L$^{-1}$ at -15 °C. Fungi were only detected in the sample during the storm and not the flanking periods.

The airmass compositions were variable, with maxima of 100% ice, 64% ice-free ocean, 33% land, and 8% MIZ over the event, indicating combined terrestrial and marine influences.

**3.4 The regional nature of Arctic INPs**

The concentrations and seasonal cycle of INPs observed during MOSAiC agreed well with mesurements at Zeppelin

Observatory (474 m above sea level) in Svalbard (Fig. 7; (Pereira Freitas et al., 2023; Tobo et al., 2024)), suggesting a commonality of INP concentrations in this Arctic region. Some differences were expected with averaging times: the Zeppelin filters were collected over one week and might miss short-term variations. Mean concentrations at -15 °C differed between the sites only by a factor of around 2 in the fall, winter, and spring, but, at MOSAiC, were as much as an order of magnitude higher in late June and early July (up to 1.4 L$^{-1}$ at -15 °C). This enhancement could indicate local influence, which agrees with the

higher ice/melt pond source attribution of ITS data during this time (Fig. 4 and 5). For activation temperatures of -25 °C, which can be less influenced by biogenic sources (Fig. 2), concentrations were more similar over the year. We note Figure 7 indicates much higher INP concentrations than total size-resolved measurements in Creamean et al. (2022): see Text S1.





**Figure 7:** INP concentration time series during the MOSAiC campaign at -10 °C, -15 °C, -20 °C, and -25 °C. CRAFT (green) refers to data taken at Zeppelin Observatory at Svalbard (Pereira Freitas et al., 2023; Tobo et al., 2024); IS (purple) refers to polycarbonate filter samples analyzed with the Ice Spectrometer.

## 4 Conclusions

The bioaerosol annual cycle in the central Arctic had a highly variable bacterial and fungal composition. The diversity of taxa, and source attribution using 16S and ITS aerosol, suggested a mixture of bioaerosols from local and distant sources. Long-range transport episodes were clearly indicated by mixed taxa and increased INP concentrations during WAMI events.

Throughout the annual cycle and across all temperatures, organic (predominantly heat-labile) INPs constituted large fractions of the INP population. These observations led to the surprising conclusion that biological INPs were present year



round, and dominated the entire INP temperature spectrum in summer. This latter point was further reinforced by the 100-fold

increase in INPs active at -15 °C, compared to the concentrations during the rest of the year. The frequency of detection of local Arctic fungal and bacterial tracers also increased in summer, and fungal source tracking identified sea ice and melt ponds as potential sources, in addition to fungi likely from local terrestrial sources with greater relative abundance of Basidiomycota. These observations pointed to increased local contributions to the bioaerosol and, by inference, to the INP population. However, the presence of likely biological INPs throughout the year was unexpected, and may be linked to the presence of

fungi throughout most of the year.

INP concentrations were largely similar at MOSAiC and Zeppelin Observatory (nearly identical below -20 °C), despite being separated by 300-1600 km (mean=1000 km) horizontally and 500 m vertically. This suggests these sites sampled a regional-scale INP population arising from mixed sources throughout the year, and unlikely from point sources proximate to either location, as concluded from the bioaerosol data. Warm-temperature INPs also increased at both sites in summer in

response to increased marine biological activity and strong terrestrial sources from decreased snow cover.

In general, these findings are relevant for Arctic mixed-phase clouds, as they exist between -25 and -5 °C and are present throughout the year (greatest fractions in spring and fall). The INPs active at temperatures that were observed during MOSAiC in spring and fall had relatively low number concentrations compared with those in summer. As the Arctic warms, the enhanced INP concentrations associated with summertime biological activity could expand into late spring and early fall, changing the

annual cycle of INPs and bioaerosols, and potentially impacting cloud properties. These data provide an important baseline for evaluating how interannual variability and longer-term trends in Arctic climate manifest in the composition and loading of airborne microorganisms, INP concentrations, and cloud properties.

**Data availability**

DOI for the DNA data has been submitted and processed to NCBI under submission ID: SUB14287695. INP data for MOSAiC is published at https://doi.org/10.5439/1804484 and for the Zeppelin Observatory in the Arctic Data archive System (ADS) at https://ads.nipr.ac.jp/data/meta/A20230821-002.

**Supplement link**

TBD

**Author contribution**

JMC, SMK, PJD, and TCJH conceptualized the sampling campaign. KRB and TCJH processed the samples. KRB performed the sample analysis and wrote the manuscript with contributions from all coauthors.

**Competing interests**

The authors declare that they have no conflict of interest.

**Acknowledgements**



This work was carried out, and data used in this manuscript were produced, as part of the Multidisciplinary drifting Observatory for the Study of Arctic Climate (MOSAiC20192020). The authors would like to thank all persons involved in the expedition of the Research Vessel Polarstern during MOSAiC in 2019–2020 (AWI_PS122_00). An extended MOSAiC acknowledgement is given in Nixdorf et al. (2021). Special thanks are given to Jeff Bowman, Emelia Chamberlain, Julia Schmale, Tuija Jokinen,

Matthew Shupe, Christian Pilz, Zoe Brasseur, and Tiia Lauila for collecting the source samples and to the staff of the Norwegian Polar Institute for their assistance with year-round measurements at the Zeppelin Observatory. This work was supported by the U.S. Department of Energy Atmospheric Radiation Measurement facility (DE-AC05-76RL01830), Atmospheric Systems Research program (DE-SC0022046, DE-SC0019745), JSPS KAKENHI (JP19H01972, JP24H00761), the Arctic Challenge for Sustainability II (ArCS II) Project (JPMXD1420318865), and the Environment Research and

Technology Development Fund (JPMEERF20172003, JPMEERF20202003, JPMEERF20232001) of the Environmental Restoration and Conservation Agency of Japan.

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
