# Peer review of "Bioaerosols as indicators of central Arctic ice nucleating particle sources"

_EGUsphere, 2025_

## Author Comment (AC1)

Reviewer responses are given in blue.

The study presented by Barry et al. investigates the seasonal cycle of INP concentration and bioaerosols based on measurements taken onboard the research vessel Polarstern over a one-year period. They find significant variability in the bacterial and fungal composition, suggesting a mixture of local, regional, and long-range transported bioaerosols. Moreover, the authors state that biological particles contribute significantly to the INP population throughout the year, dominating it in summer.

The study is very well written and presents an interesting dataset from a unique campaign. However, my major concern is about the methods used to infer information about the biological, organic, and inorganic content of the INP samples using heat treatments and $H_2O_2$ digestion, and consequently, how the results of these treatments are discussed. I am aware that such treatments are frequently used nowadays to investigate contributing species to the INP population. However, recent studies have shown that wet heat treatments can also alter the ice nucleation ability of some mineral particles, while some biogenic INPs are not affected by heat treatments (Daily et al., 2022). Using this method alone to infer contributions from biological aerosols to INPs is therefore not sufficient

Response to Major concern: We address this concern in detail in the response below to the question about Figure 2. We have also added text to the manuscript accordingly to explain prior tests reported to confirm the interpretation of the treatments.

Minor

Thank you to Reviewer 1 for the helpful responses to improve this manuscript. We have addressed all concerns and changed the text and figures to have greater clarity.

Abstract: More information about the measurement methods could be given here (e.g., how INPs were measured, the temperature range and the time resolution of the measurements.

We added: "with 3-day filters for amplicon sequencing and cumulative INP concentrations from -5 to -30 °C." (Line 17)

Line 55: „campaign" is double.

Thank you, this is corrected.

Lines 62 – 64: Given that the abundance of INPs and bioaerosols might be different depending on the sampling location within the Arctic, a map of the legs could be helpful.

We have added the combined ship track for the period of time when samples were collected, that is now Figure S1.

"The other periods were time when the ship was in transit, but we include collected samples between October 27, 2019-September 25, 2020 (Figure S1)." Lines (65-66)

[Figure]

Line 74: With a three-day time resolution of the filter samples, I assume that impact from the reserach vessel itself can occur. How could this impact the results of the INP and DNA analysis?

For INPs, the effect of the research vessel from soot contamination is likely the main concern (soot grayness will also correlate with general level of stack contamination), but we did a test showing that two filters with different degrees of soot contamination (one slight and the other moderate) collected at the same time of year had virtually identical INP concentrations (i.e., the additional soot loading did not increase INPs). It has been shown that soot is a poor INP in the immersion mode (Kanji et al. 2020). For DNA, we are mostly concerned with biological contamination from the ship, as common human contaminants, such as E. coli, exist. For those we had proceeded with a rigorous blank ASV removal algorithm that is detailed in Lines 132-137. It will be impossible to remove all contaminants for either INP or DNA, but we believe we have controlled for this the best we can, without artificially influencing the samples.

Kanji, Z. A., Welti, A., Corbin, J. C., & Mensah, A. A. (2020). Black carbon particles do not matter for immersion mode ice nucleation. Geophysical Research Letters, 46, e2019GL086764. https://doi.org/10.1029/2019GL086764

[Figure]

Line 178: As part of the data is presented in Creamean et al. (2022), it might be worth to mention the difference between this and their study.

We have added details: "First, we present the seasonal cycle of INP composition for MOSAiC, partially presented in the Creamean et al. (2022) supplement to add context to their size-resolved data." Lines (180-181)

Line 182: Are there measurement or modeling information about the Arctic haze occurrance during this year? And does it align well with the Fig. 2?

The Arctic haze season peaked earlier during the MOSAiC season, which generally aligns with Figure 2.

We have added: "The Arctic haze season during MOSAiC was stronger and peaked earlier than normal (January and February), with a large positive Arctic Oscillation phase, and which could have contributed the large inorganic INP fractions seen during this time colder than -20 °C (Boyer et al., 2023)." Lines (188-190)

Boyer, M., Aliaga, D., Pernov, J. B., Angot, H., Quéléver, L. L. J., Dada, L., Heutte, B., Dall'Osto, M., Beddows, D. C. S., Brasseur, Z., Beck, I., Bucci, S., Duetsch, M., Stohl, A., Laurila, T., Asmi, E., Massling, A., Thomas, D. C., Nøjgaard, J. K., Chan, T., Sharma, S., Tunved, P., Krejci, R., Hansson, H. C., Bianchi, F., Lehtipalo, K., Wiedensohler, A., Weinhold, K., Kulmala, M., Petäjä, T., Sipilä, M., Schmale, J., and Jokinen, T.: A full year of aerosol size distribution data from the central Arctic under an extreme positive Arctic Oscillation: insights from the Multidisciplinary drifting Observatory for the Study of Arctic Climate (MOSAiC) expedition, Atmos. Chem. Phys., 23, 389–415, https://doi.org/10.5194/acp-23-389-2023, 2023.

3: The marine impact in January is quite high as compared to the other winter months, is there an explanation for it?

Blowing snow is one explanation, as there were several events during January (Bergner et al. 2025) that aligned with a high abundance of marine taxa, especially collected in the January 15-18 filter. Particles in the snow could sublimate in the atmosphere. Transport of marine airmasses in the Arctic Haze is also possible, such as from deposition of sea spray and re-emission during blowing snow periods.

Nora Bergner, Benjamin Heutte, Ivo Beck, Jakob B. Pernov, Hélène Angot, Stephen R. Arnold, Matthew Boyer, Jessie M. Creamean, Ronny Engelmann, Markus M. Frey, Xianda Gong, Silvia Henning, Tamora James, Tuija Jokinen, Gina Jozef, Markku Kulmala, Tiia Laurila, Michael Lonardi, Amy R. Macfarlane, Sergey Y. Matrosov, Jessica A. Mirrielees, Tuukka Petäjä, Kerri A. Pratt, Lauriane L. J. Quéléver, Martin Schneebeli, Janek Uin, Jian Wang, Julia Schmale; Characteristics and effects of aerosols during blowing snow events in the central Arctic. *Elementa: Science of the Anthropocene* 3 January 2025; 13 (1): 00047. doi: https://doi.org/10.1525/elementa.2024.00047

Line 180: Similar to the explanation of „heat labile (presumably biological)", an explanation to heat stable organics and inorganics might help the reader to put this in context.

Thanks for the suggestion, we have added: "heat stable organics (e.g. from soil dust or sea spray), and inorganic (presumably mineral) (Fig. 2)." (Lines 182-183)

Figure 2: Sample Aug 2 seems to be different to the other summer samples, any explanation for it?

Yes, one potential explanation was found in the back trajectory analysis. We have added: "The August 2-5 filter was marked by lower concentrations and a lower fraction of heat labile INPs at -25 °C, which may have resulted from an airmass transition sampling air from predominantly over the ocean to over the sea ice (Fig. 6)." (Lines 184-186)

Figure 2: It is interesting to see that at T -25 °C in summer, also heat labile (biological) INPs are dominating the INP population, as this is a temperature range where mostly dust particles contribute to the INP population. Thus I am wondering if the treatments are really giving information about the inorganic or biological content (see my major concern). Are there other studies suggesting that dust (e.g., from local sources such as glacial dust) are not as important during this season? It is interesting as emissions of Arctic dust is largest in late spring summer, early autumn, depending on location (e.g., Bullard, 2012; Groot Zwaaftink et al., 2016). Is there any information about the abundance of dust particles during the here presented measurement period?

Thanks for your comment. There is one MOSAiC study, Ansmann et al. 2023, which found lidar retrievals of dust fraction of less than 5% throughout the MOSAiC year. We saw evidence of terrestrial influence at all times of year, despite a greater marine influence in the summer. Many of the terrestrial particles in summer can come from Arctic and glacial soil dust, so we are not suggesting that the INP contribution from this source is negligible. However, from previous work, we think that the main driver of the high INP activity in summer, even at colder

temperatures, is attributed to the organic component instead of their mineral component. This has been shown through hydrogen peroxide digestion by Tobo et al. 2019.

Additionally, we have recently done a multi-enzymatic digestion comprising 6 cell wall enzymes (targeting bacteria, filamentous fungi and yeast) followed by a general protease to digest the liberated proteins and the proteins within them on an aliquot of a summer MOSAiC sample to show results comparable to those obtained with the 95 °C treatment. The 95 °C treatment has been done for samples for a variety of previous environments (McCluskey et al., 2018; Suski et al., 2018; Barry et al., 2021; Knopf et al., 2021; Testa et al., 2021, DeMott et al. 2025). We believe it provides a reasonable estimate of biological INPs in this location, and the best tool available at this point in time.

[Figure]

We will add this to Figure S9, and in the main text: "Additionally, we compared the difference between 95 °C heating and an enzymatic digestion on a summer sample (Figure S9) and found them to be comparable." (Lines 195-196)

As regards to the impact of $H_2O_2$ digestion likely effectively targeting inorganics such as mineral dust, evidence presented in Tobo et al. (2014), Hill et al. (2016), Suski et al. (2018) and DeMott et al. (2025) demonstrates equivalent impacts of heated peroxide digestion on the INP activity of bulk arable soils as for the use of dry thermal treatment at 300°C, and no impact of either method on some natural desert soils. Daily et al. (2022) found little impact of dry heating on mineral particles. Furthermore, the equivalency of the wet/heated peroxide treatment of filter collected

ambient particles and dry 300°C treatment of free-flowing single ambient particles over the same collection periods in Suski et al. (2018) and DeMott et al. (2025) for two different environments suggests that the changes seen in INP spectra after treatment are due to impacts on organic INP components, leaving inorganics as the remnant INP contribution.

Ansmann, A., Ohneiser, K., Engelmann, R., Radenz, M., Griesche, H., Hofer, J., Althausen, D., Creamean, J. M., Boyer, M. C., Knopf, D. A., Dahlke, S., Maturilli, M., Gebauer, H., Bühl, J., Jimenez, C., Seifert, P., and Wandinger, U.: Annual cycle of aerosol properties over the central Arctic during MOSAiC 2019–2020 – light-extinction, CCN, and INP levels from the boundary layer to the tropopause, Atmos. Chem. Phys., 23, 12821–12849, https://doi.org/10.5194/acp-23-12821-2023, 2023.

Tobo, Y., Adachi, K., DeMott, P. J., Hill, T. C. J., Hamilton, D. S., Mahowald, N. M., Nagatsuka, N., Ohata, S., Uetake, J., Kondo, Y., & Koike, M. (2019). Glacially sourced dust as a potentially significant source of ice nucleating particles. Nature Geoscience, 12(4), 253–258. https://doi.org/10.1038/s41561-019-0314-x

McCluskey, C. S., Ovadnevaite, J., Rinaldi, M., Atkinson, J., Belosi, F., Ceburnis, D., Marullo, S., Hill, T. C. J., Lohmann, U., Kanji, Z. A., O'Dowd, C., Kreidenweis, S. M., & DeMott, P. J. (2018). Marine and Terrestrial Organic Ice-Nucleating Particles in Pristine Marine to Continentally Influenced Northeast Atlantic Air Masses. Journal of Geophysical Research: Atmospheres, 123(11), 6196–6212. https://doi.org/10.1029/2017JD028033

Suski, K. J., Hill, T. C. J., Levin, E. J. T., Miller, A., DeMott, P. J., & Kreidenweis, S. M. (2018). Agricultural harvesting emissions of ice-nucleating particles. Atmospheric Chemistry and Physics, 18(18), 13755–13771. https://doi.org/10.5194/acp-18-13755-2018

Barry, KR, Hill, TCJ, Levin, EJT, Twohy, CH, Moore, KA, Weller, ZD, Toohey, DW, Reeves, M, Campos, T, Geiss, R, Fischer, EV, Kreidenweis, SM, DeMott, PJ. 2021b. Observations of ice nucleating particles in the free troposphere from western US wildfires. Journal of Geophysical Research: Atmospheres 126: e2020JD033752. DOI: https://doi.org/10.1029/2020JD033752.

Knopf, D. A., and Coauthors, 2021: Aerosol–Ice Formation Closure: A Southern Great Plains Field Campaign. *Bull. Amer. Meteor. Soc.*, **102**, E1952–E1971, https://doi.org/10.1175/BAMS-D-20-0151.1.

Testa, B., Hill, T. C. J., Marsden, N. A., Barry, K. R., Hume, C. C., Bian, Q., Uetake, J., Hare, H., Perkins, R. J., Möhler, O., Kreidenweis, S. M., & DeMott, P. J. (2021). Ice Nucleating Particle Connections to Regional Argentinian Land Surface Emissions and Weather During the Cloud, Aerosol, and Complex Terrain Interactions Experiment. Journal of Geophysical Research: Atmospheres, 126(23). https://doi.org/10.1029/2021JD035186

Paul J. DeMott, Benjamin E. Swanson, Jessie M. Creamean, Yutaka Tobo, Thomas C. J. Hill, Kevin R. Barry, Ivo F. Beck, Gabriel P. Frietas, Dominic Heslin-Rees, Christian P. Lackner, Julia Schmale, Radovan Krejci, Paul Zieger, Bart Geerts, Sonia M. Kreidenweis; Ice nucleating particle sources and transports between the Central and Southern Arctic regions

during winter cold air outbreaks. *Elementa: Science of the Anthropocene* 3 January 2025; 13 (1): 00063. doi: https://doi.org/10.1525/elementa.2024.00063

Hill, T. C. J., DeMott, P. J., Tobo, Y., Fröhlich-Nowoisky, J., Moffett, B. F., Franc, G. D., & Kreidenweis, S. M. (2016). Sources of organic ice nucleating particles in soils. Atmospheric Chemistry and Physics, 16(11), 7195–7211. https://doi.org/10.5194/acp-16-7195-2016

Daily, Martin I.; Tarn, Mark D.; Whale, Thomas F. An evaluation of the heat test for the ice-nucleating ability of minerals and biological material Atmospheric Measurement Techniques, Vol. 15, Issue 8 https://doi.org/10.5194/amt-15-2635-2022

Section 3.2: Are there studies about the ice nucleation activity of the discussed bacterial taxas?

Ice nucleation has been attributed primarily to three genera: *Pseudomonas, Pantoea, and Xanthomonas* (Hill et al. 2018). These genera were not associated with our top-20 found during MOSAiC. There are still a lot of unknowns about biological ice nucleation. Even though we have evidence for the INPs being of biological origin, we cannot provide a definite link with the amplicon sequencing information here, and the bacterial INPs will come from a small percentage of the total bacteria. The IN activity could also be attributed to fungi.

Hill, T. C. J., DeMott, P. J., Conen, F., & Möhler, O. (2018). Impacts of Bioaerosols on Atmospheric Ice Nucleation Processes. In A.-M. Delort & P. Amato (Eds.), Microbiology of Aerosols (1st ed., pp. 197–219). John Wiley & Sons.

Section 3.2: How does the seasonal cycle of bioaerosol relate to the Arctic haze phenomena?

We will add: "The seasonal cycle of the Arctic haze corresponded to the occurrence of a diverse population of bioaerosols that was most likely from longer range transport, with a lower population of marine taxa." (Lines 221-223)

Figure 3 and Figure 5: Why do these figures have headers?

Thank you for pointing this out, they are now removed.

Lines 247 – 248: Only when reading the figure caption of Fig. 6 it became clear to me what you mean with the relative percentages of each zonal coverage, I suggest to explain it in more detail in the text.

We have changed the text to closely match the caption:

"Ice is defined as greater than 85% sea ice concentration (SIC), marginal ice zone (MIZ) is 15-85% SIC, ocean is <15% SIC, and land." (Lines 257-259)

---

## Author Comment (AC2)

Reviewer responses are given in blue.

In this study, filters collected during the MOSAiC cruise in the Arctic were examined. DNA analysis was done to examine the present bioaerosols, together with measurements of INPs (ice nucleating particles). The data was amended with data from filters from other Arctic samples such as seawater, sea ice, snow etc. . Backward trajectories were also analyzed to zero in on the sources of bioaerosols and INPs.

As results, the by now well-known seasonal cycles of INP concentrations were presented, together with seasonal cycles for bioaerosols. Also, some ideas on possible sources, specifically for the bioaerosols, were given. For the latter, both long-rang transport as well as local influence was observed. While some terrestrial influence on the bioaerosols was seen throughout the year, in the summer month most bioaerosols clearly came from marine sources. Nevertheless, the fungal contributions predominantly point to terrestrial sources. Also, a comparison to INP concentrations from Svalbard shows closeness of the data for most of the time, which is interesting given that the distance between the ship and Svalbard was varying and sometimes large.

This all adds nice bits and pieces to things the community already understood about bioaerosols and INPs, and certainly merits publication. The methods are all sound, the text is well structured and well written. It only occurred to me if this is not better published as a measurement report rather than a full scientific publication. But this is a decision the editor should make.

There is one really unsettling information in the manuscript, which is the discrepancy between the INP concentrations published in Creamean et al. (2022) and in here, which both come from samples collected onboard the Polarstern simultaneously. I suggest below to include a comparison with other Arctic data to learn which of the two datasets may be closer to these. This may be included in the main text or the SI. But it should be done.

As the number of my comments and remarks below is rather small, they are all just given one after the other. And, as said, publication can certainly be granted once these few small and the one larger issue are taken care of.

Thank you to Reviewer 2 for the helpful responses to improve this manuscript. We have made all suggested changes, added clarification, and improved the discussion surrounding the measurement discrepancy. We address the discrepancy between Creamean et al. (2022) and our findings in the response below.

Comments:

Line 185-186: There is a "," missing between "productivity" and "sea". - Also, you mention "less snow coverage" – where was that (as you were on a ship)? Are you referring to snow on land? Please clarify.

The text now reads: "The heat labile maximum in summer is reflective of enhanced biological productivity, sea ice minimum, glacial retreat, and less terrestrial snow coverage." (Lines 192-193)

Line 209: Change "wasn't" to "was not".

The text now reads: "The alpha diversity of the aerosol samples was not significantly ($p<0.05$) different between seasons (Fig. S5), despite increased variability in the spring and summer, providing further evidence that the bioaerosols were diverse taxonomic mixtures." (Lines 217-219)

Line 275: Please add a number for what you consider are "higher INP concentrations".

Thanks for the suggestion. The text now clarifies the value in reference to the filter collected during November 18-21: "After November 18-21, *Polaribacter* was not detected again until May 2020, and higher INP concentrations at -15 °C ($>1.8 \times 10^{-3}$ L$^{-1}$) were not detected until January." (Lines 287-288)

Line 289: Add to this sentence where the Polarstern was, compared to Svalbard, in June and July, or at least point to Fig. 6 where the distance between both can be seen.

Thank you, the text now reads: "Mean concentrations at -15 °C differed between the sites only by a factor of around 2 in the fall, winter, and spring, but, at MOSAiC, were as much as an order of magnitude higher in late June and early July (up to 1.4 L-1 at -15 °C), when the *Polarstern* was an average of 450 km away from Svalbard (Fig. 6)." (Lines 298-301)

Line 291-292: The mentioned difference between your two datasets (one published earlier and the one included here) is unsettling. In the previous part you argue that INP concentrations quite similar between Svalbard and the Polarstern, no matter the distance between the two. And it is likely adequate to assume a somewhat repeatable annual cycle in INP concentrations. There were other Arctic long-term data published before, for land-based INP sampling, connected to parameterizations (e.g., Li et al., 20; Sze et al., 2023; the latter having data taken simultaneously to your measurements in northern Greenland). It would be very instructive to compare your data with these and see if either of your datasets matches, and which one. Please add this comparison to your manuscript or to the SI.

Thank you for the suggestions. We have added one new supplemental figure (Figure S8), adopting the suggestion to additionally compare to Li et al. 2022 and Sze et al. 2023 as they were both collected in the same time frame as MOSAiC. Additionally, we include another figure, a measurement cooling slope comparison figure in this response. Although we don't expect the concentrations to match entirely since they were collected near ground level and near land, these two added studies are still much closer to the polycarbonate filters collected on the *Polarstern* than the DRUM. In addition, we also added to the end of Text S1: "This different cooling rate was a function of some DRUM samples (primarily spring and summer) being analyzed in a different laboratory with a ~15-20 °C colder ambient temperature. The sample integration time differences (24- versus 72-hour) could result in short term concentration discrepancies but should not influence the seasonal averages. Comparisons with two other Arctic ground studies that made

measurements at the same time as MOSAiC (Li et al., 2022; Sze et al., 2023) revealed much better agreement with the polycarbonate filters analyzed with the IS (Fig. S7), which gives confidence in this dataset as representative of both the central and wider Arctic basin. Comparison tests and investigations of this unexpected result are ongoing. When DRUM samples were run on the IS, the agreement was generally much better to the total aerosol filters, suggesting measurement techniques account for at least some of the discrepancies (Fig. S7). Overall, the INPs from the DRUM and cold plate analysis should be viewed as a lower bound or subset of observed total aerosol INPs. The use of only the total aerosol INPs for future studies is recommended as it is more representative."

[Figure]

[Figure]

SI, line 20: Sampling time typically is taken care of when INP concentrations are calculated. That parameter should not show up in seasonal average values, unless there would be clogging of the filters for the longer sampling times, which, given the environment in which you took your samples, is unlikely. Remove the "sample integration time" from your list of possible differences, and maybe explain in a separate sentence that this should not influence the results if measurements are done properly.

We agree and have removed that from the sample integration time, and added the clarification sentence: "The sample integration time differences (24- versus 72-hour) could result in short term concentration discrepancies but should not influence the seasonal averages." (Lines 24-26)